# Melanoma Mediated Disruption of Brain Endothelial Barrier Integrity Is Not Prevented by the Inhibition of Matrix Metalloproteinases and Proteases

**DOI:** 10.3390/bios12080660

**Published:** 2022-08-19

**Authors:** Akshata Anchan, Graeme Finlay, Catherine E. Angel, James J. W. Hucklesby, Scott E. Graham

**Affiliations:** 1Department of Molecular Medicine and Pathology, School of Medical Sciences, Faculty of Medical and Health Sciences, University of Auckland, Auckland 1023, New Zealand; 2Centre for Brain Research, University of Auckland, Auckland 1023, New Zealand; 3Auckland Cancer Society Research Centre, University of Auckland, Auckland 1023, New Zealand; 4School of Biological Sciences, Faculty of Science, University of Auckland, Auckland 1010, New Zealand; 5Maurice Wilkins Centre for Molecular Biodiscovery, University of Auckland, Auckland 1010, New Zealand

**Keywords:** ECIS, xCELLigence, impedance, barrier function, endothelium, blood–brain barrier, melanoma, matrix metalloproteinases, proteases

## Abstract

We have previously shown that human melanoma cells rapidly decrease human brain endothelial barrier strength. Our findings showed a fast mechanism of melanoma mediated barrier disruption, which was localised to the paracellular junctions of the brain endothelial cells. Melanoma cells are known to release molecules which cleave the surrounding matrix and allow traversal within and out of their metastatic niche. Enzymatic families, such as matrix metalloproteinases (MMPs) and proteases are heavily implicated in this process and their complex nature in vivo makes them an intriguing family to assess in melanoma metastasis. Herein, we assessed the expression of MMPs and other proteases in melanoma conditioned media. Our results showed evidence of a high expression of MMP-2, but not MMP-1, -3 or -9. Other proteases including Cathepsins D and B were also detected. Recombinant MMP-2 was added to the apical face of brain endothelial cells (hCMVECs), to measure the change in barrier integrity using biosensor technology. Surprisingly, this showed no decrease in barrier strength. The addition of potent MMP inhibitors (batimastat, marimastat, ONO4817) and other protease inhibitors (such as aprotinin, Pefabloc SC and bestatin) to the brain endothelial cells, in the presence of various melanoma lines, showed no reduction in the melanoma mediated barrier disruption. The inhibitors batimastat, Pefabloc SC, antipain and bestatin alone decreased the barrier strength. These results suggest that although some MMPs and proteases are released by melanoma cells, there is no direct evidence that they are substantially involved in the **initial** melanoma-mediated disruption of the brain endothelium.

## 1. Introduction

Melanoma is an aggressive skin cancer with a high propensity to metastasise to the brain [1]. Metastasis majorly occurs via the circulation through the blood–brain barrier (BBB). Here, the brain endothelial cells form the first physical barrier that needs to be breached by blood-borne cancer cells [2,3,4]. We have recently shown using sophisticated biosensor technology, that the paracellular barrier strength of brain endothelial cells, is rapidly weakened by invasive melanoma cells [5,6]. Melanoma cells were shown to translocate rapidly to the paracellular borders of the endothelial cells, and progress to separate the junctional borders and traverse between the neighbouring endothelial cells. Importantly, the loss of barrier integrity was rapid and evident within 60 min of melanoma cell addition to the apical face of the endothelial cells. Such a fast effect suggested that melanoma cells are able to directly alter the molecular integrity of the paracellular junctional cleft. Theoretically, this can be mediated by a range of enzymes that catalyse the breakdown of proteins for various functions, such as cell and tissue remodelling, cell and tissue regulation and cell signalling, detailed below. These enzymes are proteases, which are classified into seven large families, based on their catalytic site residues, as aspartic, cysteine, serine, metallo, threonine, glutamic and asparagine peptidases [7,8]. 

One large metallopeptidase subfamily, called matrix metalloproteinases (MMPs), have been extensively studied for their importance in the melanoma metastatic cascade [9,10]. MMPs are also implicated in destabilising the neurovascular unit during intracerebral traumas [11]. MMPs are zinc-dependant endopeptidases [10,12,13] consisting at least of a pro-domain, catalytic domain and a highly conserved active site [14]. Over twenty-five different MMPs have been identified and these are categorised into different functional classes [9,14,15]. The classes define their predominant role as gelatinases, collagenases, stromelysins, matrilysins, metalloelastases, membrane-type proteases and others [16,17,18]. Cumulatively, these molecules cleave various extracellular matrix (ECM) materials, and several may interact with integrins and adhesion molecules for optimal positioning and protease activity [9]. These characteristics are essential in cancer, where ECM remodelling and regulation is a key mechanism for cancer growth, progression, and metastasis. A major cancer related MMP is the gelatinase MMP-2, which is often upregulated in melanoma [19] and is closely associated with invasion and metastasis. One of the activators of MMP-2 is another membrane-type MT1-MMP [20,21] and once activated at the membrane, MMP-2 can remain membrane bound, be released into the surrounding environment or attach to integrins [22] to degrade the ECM. The upregulation of MMP-2, with its membrane receptor at the leading front of the cell invadopodium, allows modulation of melanoma adhesion and spreading in an ECM environment [23]. In addition, MMP-2 may also bind to αVβ3 integrin to facilitate migration at the primary site [24]. MMP-2 expression is also significantly increased in the tumour tissue of patients with melanoma at the primary and secondary sites [25]. MMP-2 mRNA and protein are expressed in human melanoma mouse xenografts and this expression positively correlates with melanoma aggressiveness [26].

Other MMPs of interest include the collagenase MMP-1, stromelysin MMP-3 and gelatinase B MMP-9. MMP-1 activation is correlated with promoting melanoma progression into the more aggressive ventricle growth phase [27,28]. MMP-1 is upregulated in melanoma in vivo [26] and both MMP-1 and -3 are associated with shorter disease-free survival [29,30]. MMP-3 has also been shown to increase the BBB permeability and extravasation of dyes in mouse models [31]. Another example is MMP-9, which is suggested as a marker for treatment assessment in melanoma patients [32] and with uveal melanoma [33]. In mice, the addition of recombinant inhibitors of MMPs reduces the number of melanoma lung metastasis (but not the size of the metastasis), suggesting their importance in extravasation [34]. Collectively, this makes MMPs a crucial family to investigate in melanoma disruption of the brain endothelium.

In addition to matrix metalloproteinases, a large repertoire of enzymatic families exists as ECM degrading molecules. A serine-based protease called seprase has been closely associated with melanoma migration through the endothelial monolayer in a Transwell system [35]. Another serine protease, called kallikrein-related peptidase 6 (KLK6), was shown to increase melanoma invasion though a Matrigel scaffold when released by supporting stromal cells [36]. Additionally, cathepsins, which are a family of lysosomal proteases, are highly implicated in cancer progression and growth, and lead to poor prognosis specifically in malignant melanoma [37,38,39,40]. Cathepsins cover many different types of proteases, but the cysteine based Cathepsin B, and L and aspartic based Cathepsin D, are most associated with an increased in vitro proliferation and progression of melanoma [41,42,43]. One proposed mechanism is that Cathepsin B expression in melanoma stimulates fibroblast activation through a transforming growth factor β (TGFβ)-dependent pathway, and this stromal involvement increases melanoma traversal through the basement membrane; however, this acts only as an additive mechanism, as Cathepsin B also supports melanoma invasiveness without the presence of fibroblasts [44]. Due to their multiple and dynamic roles, it was hypothesized that other existing proteases and protease families may also be directly or indirectly correlated with cancer invasion.

This study aimed to assess the involvement of a range of proteolytic enzymes in melanoma mediated brain endothelial barrier disruption. The objective was to identify targetable proteins in the blood, prior to melanoma extravasation past the brain endothelium. The prime suspects were MMPs, which were hypothesized to facilitate the melanoma mediated loss of brain endothelial barrier strength. The theory herein, was that blocking a variety of these molecules with inhibitors of a low nanomolar potency would decrease the ability of melanoma cells to invade at the endothelial cell junctions substantially, if the proteases were majorly involved in this process. 

## 2. Materials and Methods

### 2.1. Cell Culture

#### 2.1.1. Human Brain Endothelial Cells (hCMVECs)

The human cerebral microvascular endothelial cells (hCMVECs) were an immortalized cell line, purchased from Applied Biological Materials Inc. (ABM, cat# T0259, Richmond, BC, Canada) and cultured in T75 flasks with M199 growth media containing 10% FBS, 1 µg/mL hydrocortisone, 3 ng/mL hFGF, 1 ng/mL hEGF, 10 µg/mL heparin, 2 mM GlutaMAX and 80 µM dibutyryl-cAMP (as detailed in Appendix B, Table A1). All culture-ware was coated with 1 µg/cm^2^ collagen I dissolved in 0.02 M of acetic acid to replicate a substrate of the basement membrane. In situ, the brain endothelium was polarized and had a side facing the blood (apical) and a side facing the brain parenchyma (basal side attached to basement membrane). As this paper aims to investigate melanoma-protease activity in the circulation, all treatments of the hCMVECs were conducted on the apical side of the endothelial cells which was physiologically important for our in vitro experiments. We have previously characterized these cell lines and interpreted their barrier properties on biosensors [5,45]. See Appendix A for images of hCMVECs growing in a monolayer (phase) and as stained for vascular endothelial junctional molecule VE-Cadherin (CD144) showing their cobblestone monolayer nature. 

#### 2.1.2. Malignant Melanoma Cells (NZM)

New Zealand melanoma (NZM) cells were developed from human metastatic melanomas by the Auckland Cancer Society Research Centre (ACSRC), described in [9]. Three metastatic melanoma cell lines (called NZMx), detailed in Table 1, were used. The cells were cultured with minimum essential medium α (αMEM) containing 5% FBS, 5 μg/mL insulin, 5 μg/mL transferrin and 5 ng/mL sodium selenite (as detailed in Appendix B, Table A1). As cancer cells may acquire genetic abnormalities upon multiple replications, none of the NZM lines were used past passage 30.

### 2.2. Biosensors 

Two 96 well plate type biosensors were used in this paper, the Real-time cell analysis (RTCA) xCELLigence and Electric Cell-Substrate Impedance Sensing (ECIS). We have previously shown that melanoma cells disrupt the brain endothelial barrier as detected by ECIS, which was paired with imaging data that showed gaps forming within the endothelial monolayer (Appendix A) [5]. Hence, we have established these biosensors previously for studying brain endothelial barrier formation and disruption [5,6,45,46,47,48,49] and characterized their comparative usability in Hucklesby [48]. The following Figure 1 and methods section aims to summarize our previous finding, detailed in and adapted from [5,48,50]. Note that Figure 1 shows the xCELLigence and ECIS biosensors, used in this paper. The cellZscope (cellZScope2; nanoAnalytics) was not used in this paper, but for comparison with the results of our 96 well plate array with a 3D model. Such two-chamber models are often used in the literature for the assessment of endothelial barrier function and permeability, particularly if the components of the basal side require alteration [51], and in the development of multi-cellular models of the BBB [52,53]. As our focus is primarily on the apical side of the brain endothelial cells, we commenced our experiments on our plate-based models.

#### 2.2.1. xCELLigence Theory and Setup

Real-time cell analysis (RTCA) xCELLigence (ACEA Biosciences Inc., San Diego, CA, USA) was used to measure the overall changes of the brain endothelial monolayer and inform on their adhesion properties. xCELLigence arrays are lined with gold-electrodes upon which brain endothelial cells are added. Cells that adhere to the plate surface alter the electrical impedance across the array. This impedance is recorded and converted by the xCELLigence software into a cell index (CI), which is a relative and arbitrary unit. The CI calculation is based on the following formula: CI = (Zi − Z0)/15 Ω, where Zi = impedance at the individual, experimental time points; Z0 = background impedance measured at the start of the experiment; 15 Ω = the nominal impedance value, at 10,000 Hz, which is the relevant frequency at which the current was applied. The CI is, therefore, a ratio of impedances, where the addition of cells on the electrode increases the Zi, giving a measure of the cellular adhesion to the electrodes within each well. With this measure, any stimulus that induces changes in cell morphology (size, volume, shape or spreading), cell number (proliferation or death) or movement (migration or extravasation) can be investigated. As the manufacturer recommends, cells which adhere strongly to the electrodes have a larger CI value, and those which loosely adhere have a low CI value. The xCELLigence system provides an overview of the changes in cell adhesion and viability as a measure of their attachment to the electrodes. Due to the nature of MMPs to disrupt cellular and matrix-based proteins, in this paper, xCELLigence was used to assess the overall change in endothelial adhesion, and therefore CI, to the underlying matrix substrate (collagen coat in this paper), and the subsequent electrodes.

The xCELLigence 96 well plates (E-Plate VIEW 96 PET, cat# 300600910, ACEA Biosciences Inc.) were coated with 1 µg/cm^2^ of rat-tail collagen I dissolved in 0.02 M of acetic acid as per the cell culture protocol described above. The hCMVECs were seeded at 20,000 cells per well, in 100 µL of complete M199 growth media and allowed to grow until they formed their monolayer, which typically took approximately 48 h. The formation of the monolayer was determined once the increase in the cell index plateaued and stabilised.

#### 2.2.2. Electric Cell-Substrate Impedance Sensing (ECIS) Setup

ECIS is a real-time and label-free, impedance-based method used to assess the structural integrity of cellular barriers, such as that formed by brain endothelial cells. Barrier strength is assessed by measuring the impedance across a confluent monolayer. 

**Figure 1 biosensors-12-00660-f001:**
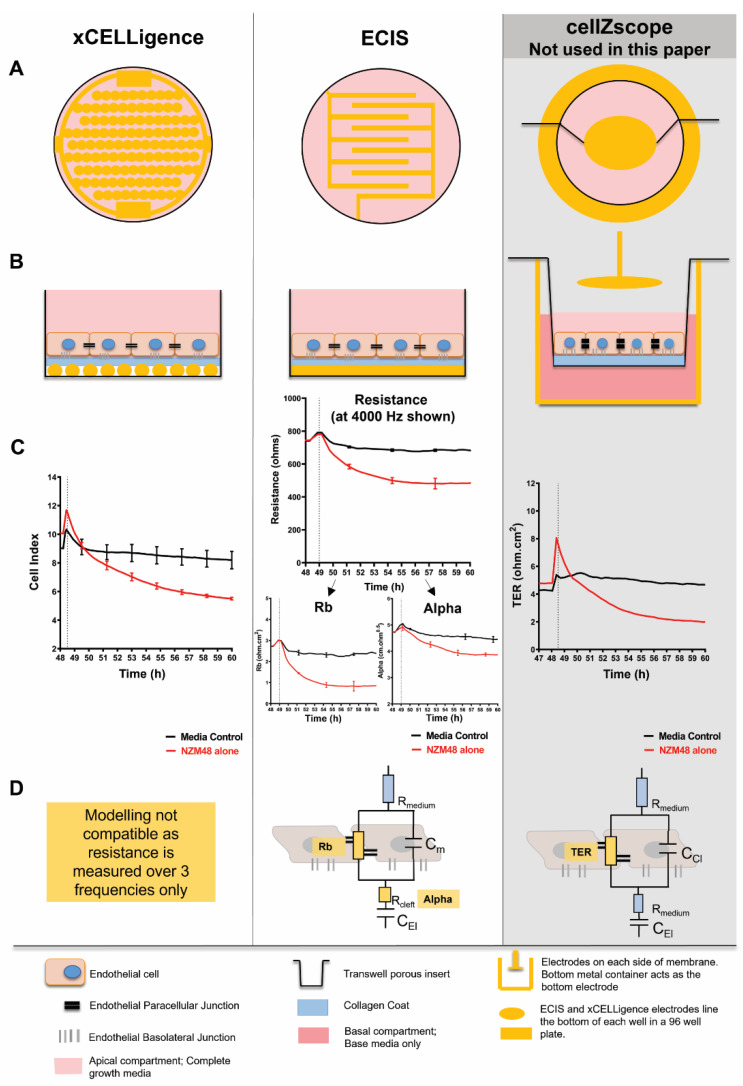
Schematic explaining biosensors used and their comparative measurements. The xCELLigence and ECIS array are shown, used in this paper. The cellZscope, a Transwell-based array is also shown as a comparison but not used in this paper. (**A**) Top-view of electrode arrangement for the ECIS (96W20idf plate), xCELLigence (E-plate) and cellZscope2 instruments. Note that ECIS and xCELLigence are 96 well plate systems. Both the ECIS and xCELLigence electrodes have a similar interdigitating electrode configuration, which covers a high proportion of the bottom of the well. The 96 well plate systems have direct contact with the endothelial cells (and substrate); however, the cellZscope does not. (**B**) Profile view of apparatus set-up and the arrangement of endothelial cells in the systems. (**C**) Typically displayed measurements for each system; xCELLigence shows the unit-less Cell Index as a measure of cell attachment to the electrode; ECIS shows the resistance at 4000 Hz which is the direct read out of the impedance across the monolayer of endothelial cells. This resistance recorded at several frequencies was modelled by the ECIS software to give the Rb (paracellular) and Alpha (basolateral) barrier strength; cellZscope shows the transendothelial electric resistance (TER) that can be determined by the cellZscope software by recording resistance across the two-chamber system across multiple frequencies. (**D**) Simplified circuit that suggests all the parameters that can be modelled by the frequency sweeps on ECIS and cellZscope.

Unlike the xCELLigence system, in ECIS changes in barrier integrity can be spatially attributed to specific areas of a cellular barrier through mathematical modelling, as detailed previously in [5,45]. Essentially, as the endothelial cells grow, the resistance across the well increases and eventually plateaus when the cells are confluent. Upon treatment with factors that disrupt the endothelial barrier, the junctions weaken, current flows more freely between the cells and the electrical resistance decreases (Appendix A). ECIS resistance is measured and recorded over several frequencies of current and this allows for the data to be modelled depending on the frequency at which the current is applied by the electrodes. At low frequencies (100–10,000 Hz), the current flows only between cells, whereas at high frequencies (>10,000 Hz), the current can also flow through cells. Collectively, the low and high frequency data can be mathematically modelled to provide information on the resistance between cells (Rb-paracellular) and resistance between cells and the underlying substrate (Alpha-basolateral) (Appendix A). ECIS data measured at one frequency of 4000 Hz reflect the closest depiction of change in the resistance caused by the cell morphology on endothelial cells, shown first by Tiruppathi [54] and as advised by the manufacturers. In this paper, we have shown the unmodelled resistance at 4000 Hz as a first overview of the change in cell-based resistance in the system. If an effect was seen in the resistance at 4000 Hz first, we interpreted, showed and modelled Rb and Alpha, to discern which aspect of the barrier was affected most.

ECIS Zθ 96 well plates (96W20idf) were treated with 10 mM cysteine for 15 min to maintain the electrode capacitance. The wells were coated with 1 µg/cm^2^ of collagen I in 0.02 M of acetic acid, following which, the hCMVECs were seeded at 20,000 cells per well, in 100 µL of complete M199 media. The ECIS machine was run continuously at multi-frequencies, so that the ECIS system could record resistance across low and high frequencies and then model the recorded resistance into separate components as developed by Giaever and Keese [55].

### 2.3. Protease Based Treatment of Brain Endothelial Cells

In all instances of the exogenous addition of commercially bought materials, the hCMVECs were previously seeded in 100 μL of complete M199 to allow their growth into confluent monolayers. Exogenous proteinases and their inhibitors were prepared in complete M199 media at 2× the effective concentration for the experiments. Amounts of 100 μL of the individual exogenous molecules were added to the hCMVECs which were in 100 μL of media, resulting in a 50% dilution to the final effective concentration for the experiments. The details of all exogenously added materials are in Appendix B, Table A2.

#### Protease Based Treatment of Melanoma Addition to Brain Endothelial Cells

Three independent NZM lines were harvested with TrypLE (cat# 12604021, Gibco, Thermo Fisher Scientific, Waltham, MA, USA) to protect any cell-surface molecules that aid melanoma cell adhesion. The TrypLE was then diluted and centrifuged out of the melanoma solution, to ensure the NZM cells were free of the dissociation reagent before treatment. Live melanoma cells were carefully counted to ascertain an effector-target (E:T) ratio of 1 melanoma cell:1 hCMVEC (1:1). All the inhibitors were reconstituted in pre-warmed media at 2× the effective concentration. The MMP inhibitors were serially diluted at 1:10 to obtain a working concentration range. The relevant inhibitors were carefully mixed with the NZM cells prior to their addition to the hCMVECs, diluting the inhibitor by 50%. The NZM-inhibitor cocktail was incubated for 5 min, after which 100 µL of NZM-inhibitor cocktail was added to the hCMVECs, pre-existing in 100 µL of media, diluting the inhibitor concentration by a further 50%. Changes in the endothelial barrier were measured autonomously using either xCELLigence to measure the endothelial cell adhesion and impedance, or ECIS to measure the specific barrier resistance and integrity. The specific details of this assay are given in Table 2.

### 2.4. Melanoma Protease Detection 

NZM cells were seeded at a density of 900,000 cells in a T75 flask. Media was collected from the cells from Day 1 (day of cell passage) to Day 7 (~90% confluency) on alternating days. The collected conditioned media was centrifuged for 10 min at 300× *g* to remove cellular debris and the supernatant was stored at −80 °C. On the day of the experiments, the samples were thawed (for single-use only) and prepared as per the manufacturer’s protocol for the respective methods below.

#### 2.4.1. Luminex Immunoassay

The expression of four critical MMPs was quantified using a Luminex Immunoassay (R&D Systems, Inc., Minneapolis, MN, USA), which is a multiplex bead-based assay that allows the testing of different secretory factors, concurrently. On the day of the experiment, a capture bead cocktail, a reporter antibody cocktail and a standard curve were prepared for each secreted protein of interest (detailed in Table A2 in Appendix B, including the std. curve range). The samples were prepared according to the manufacturer’s protocol and as detailed in [49,56]. The samples and standards were run on the Accuri C6 flow cytometer (MMP-1 non-magnetic plex) and the Luminex Magpix^®^ System (Luminex, Austin, TX, USA—MMP-2, -3, -9 magnetic plex). The results were analysed on GraphPad Prism (version 7.03, GraphPad, San Diego, CA, USA) using the generated standard curve.

#### 2.4.2. Screening Proteome Profiler Arrays

A Proteome Profiler (R&D Systems, Inc., Minneapolis, MN, USA) was used to screen the NZM conditioned media for several proteases. Two Proteome Profiler arrays were used: Human XL Cytokine Array (cat# ARY022B) and Human XL Oncology (cat# ARY026), according to the manufacturer’s protocol. The Proteome Profiler arrays were imaged using the ChemiDoc™ MP (Bio-rad, Hercules, CA, USA) and ImageLab v6.0.1 software (Bio-rad, Hercules, CA, USA), at an exposure time of 10 s, with signal accumulation. Chemiluminescence spots were analysed to find the first evidence of saturation on the reference spot, and the image collected 120 s after this was used for analysis using an ImageJ v1.50i blot analysis.

### 2.5. Biosensor Statistics

RStudio (version 1.1.414, RStudio, Inc., Boston, MA, USA) was used to conduct a two-way analysis of variance followed by a Tukey’s range test on the biosensors data. All the probabilities shown are relative to a media-only control at the final time point, unless otherwise stated. Normality was confirmed using both a visual inspection of the data and the Shapiro–Wilk test of normality. All the graphs were generated using GraphPad Prism (version 7.03). The R package for this analysis is being written into a package for publishing and open access use [5,6,48].

## 3. Results

### 3.1. MMP-2 Is Expressed in Melanoma Conditioned Media but Does Not Disrupt the Brain Endothelial Barrier

MMP expression was measured in the conditioned media of various melanoma lines over time, where Day 1 was the day of cell seeding, and Day 7 represented 90% confluency. The heatmap in Figure 2A shows that only MMP-2 was detected in all the melanoma lines. There was some evidence of MMP-1 expression in media collected from NZM74 on Days 7 and 5. Notably, the concentration of detected MMP was sequentially higher in the media collected from the later days. This data suggested that the MMPs did not degrade in the timeframe of the assay and their detection was evidently highest in the media collected from Day 7; therefore, conditioned media collected from Day 7 was used for all further experiments. Experimental repeats of MMPs-1, -3 and -9 showed that their expression was consistently low or undetected. MMP-2 was the only protease that was abundant in the conditioned media from all three melanoma lines and expressed at concentrations as high as 200–300 ng/mL in NZM7 (Figure 2B).

Due to this high expression and the impact of MMP-2 in the corresponding literature, the next logical step was to assess if MMP-2 directly disrupted the brain endothelial barrier and if so, investigate the molecular machinery used for this process, focusing on the integrin expression of the endothelial cells. MMPs have well established functions in regulating matrix associated proteins; therefore, commercially sourced active MMP-2 (62 kDa, [13]) was added to the brain endothelial monolayer, using the xCELLigence system to detect the **overall** impedance changes across the endothelial monolayer. Recombinant Human MMP-2 (#420-02, PeproTech, Cranbury, NJ, USA) was added to the brain endothelial monolayer in a 1:10 dilution series. A top concentration of 250 ng/mL was used to replicate (i) detected levels in the melanoma conditioned media and (ii) clinically tested, variable serum concentrations of MMPs [57,58]. MMP-2 was added directly to the apical face of the endothelial cells to target the location where the melanoma cells would first make contact. This was important to avoid any direct MMP-based degradation of the basolateral or “matrix-like” adhesions, which would not be possible with an intact barrier.

Surprisingly, there was no change in the endothelial cell index (impedance) upon an addition of MMP-2 at any of the concentrations (Figure 2C). This suggested that MMP-2 alone did not cause the direct disruption mediated by the melanoma cells, even though there was evidence that it was most abundantly expressed. We then inquired if perhaps other MMPs and also other proteases were rapidly released only post co-culture with the brain endothelial cells, and that there was in fact a cohort of proteases working together. As it is difficult to measure protein expression only by the melanoma cells in a co-culture system, we decided the next reasonable step was to block *all relevant endogenous* MMP activity occurring during melanoma mediated disruption of the brain endothelial barrier. This was performed to take a broader approach and assess the effect of broad-spectrum MMP inhibitors on a melanoma mediated disruption of the brain endothelial. Herein if an effect was seen, then the idea was that at least one if not more of the inhibited MMPs play an important role in facilitating melanoma disruption of the brain endothelial barrier integrity. The same strategy was used for several other proteases of interest later in this paper. The following assay was conducted using commercially available inhibitors, with known target specificity, especially to MMP-2 (Table 3–Starred*). Three different synthetic broad-spectrum inhibitors were used to cover a large range of MMPs. The IC_50_ values for every MMP inhibited was different and, therefore, each inhibitor was added at a very high 5 µM top concentration and then applied in a 1:10 dilution series. 

### 3.2. Blocking Melanoma Cells with MMP Inhibitors Does Not Prevent Melanoma Mediated Brain Endothelial Disruption

The results were analysed to identify if any of the MMP inhibitors substantially inhibited the brain endothelial barrier disruption caused by the melanoma addition. Note that we were interested in the initial response caused by the melanoma cells, which showed disruption occurring within the first few hours of their addition. As an experimental cut-off, any reduction in endothelial disruption over the arbitrary 25% line (red line in Figure 3) was interpreted as a biologically significant effect. The idea herein was that overcoming this line as suggested in Appendix A, shows a substantial effect of the inhibitors in blocking disruptive melanoma proteases, thereby suggesting their importance in the observed *rapid* brain endothelium barrier disruption. Figure 3 shows that across all melanoma lines and all inhibitors, there was no substantial change in endothelial disruption. With NZM48, all the inhibitors showed a very small change in barrier disruption at the lowest drug dose of 5 nM. This was most visible with ONO4817 and was only observed for NZM48. Statistical analyses comparing NZM lines incubated with an MMP inhibitor against the NZM with a vehicle control showed all treatment variables to be statistically insignificant, even at the much later endpoint of 80 h, further supporting the null hypothesis. Most importantly, the effect of ONO4817 was seen 6 h after treatment, which is well after the initial and predominant melanoma mediated barrier loss, marked by the grey boxes.

Intriguingly, the inhibitors alone disrupted the endothelial barrier at the top concentration of 5 µM. Marimastat alone decreased the endothelial cell index in one of six experiments (three shown in Figure 3), whereas batimastat alone decreased the endothelial cell index in every experiment. In fact, the disruption by the inhibitor alone was the only (unexpected) treatment that was statistically significant from the control at the experiment endpoint. This was a curious observation and to assess this further, the inhibitors were added to the hCMVEC brain endothelial cells on our more sensitive biosensor ECIS [48,50], to assess which parameter of the endothelial barrier was affected most. The results showed that batimastat and marimastat affected both the paracellular barrier (Rb) and to a lesser extent, the basolateral barrier (Alpha), indicating that the inhibitors alone decreased the endothelial junctional barrier integrity. This was measured at a high concentration of 5 µM. Note that this occurred over a long-time frame taking up to 90 h post addition to cause a loss of paracellular barrier resistance by approximately 50% (Appendix A). As we noticed an effect of the inhibitor alone that could be detrimental to the health of the brain endothelial cells, we decided to use ECIS for all future experiments. This was undertaken to more specifically discern the barrier disruption to its Rb or Alpha parameters, if a change in the barrier resistance was evident at a resistance measured at 4000 Hz.

### 3.3. Cathepsin D, Cathepsin B and uPAR Is Expressed in Melanoma Conditioned Media, as seen with MMP-2

The results suggested that across a wide concentration range, 5 µM to 5 nM, MMP inhibitors did not substantially hinder the melanoma-mediated disruption of the brain endothelial cell barrier. In addition to matrix metalloproteinases, a large repertoire of enzymes exists that function as proteolytic regulators of their environment; hence, the next step was to assess the expression of other cancer related proteases in melanoma conditioned media. The expression was assessed using the proteome profiler, cytokine, and oncology-based kits for the large-scale measurement of proteases in three melanoma lines, concurrently. Figure 4 shows the duplicate blots of the expressed proteases, paired with the positive reference, a negative and the undetected MMP-3. Three proteases were reliably detected; the aspartic endo-protease- Cathepsin D; the lysosomal cysteine- Cathepsin B; urokinase plasminogen activator receptor- uPAR (Figure 4, Table 4). Unsurprisingly, both Cathepsin D and Cathepsin B are associated with malignant melanoma progression in human and mouse models [41,43]. 

Semi-quantitative analysis of the data showed that these three proteases, along with MMP-2 were detected at substantially higher levels than the αMEM media control. The MMPs were added to this analysis to compare with the quantitative Luminex data and as per Figure 2, MMP-2 was the only MMP reliably detected in all three melanoma lines. A range of other proteases such as Cathepsin S, complement Factor D and members of the Kallikrein protease family were also included in the proteome profiler arrays, however, none of these were detectable in the melanoma conditioned media. Two proteins of the serpin (serine protease inhibitor) family were also assessed but not detected in the conditioned media (Table 4).

### 3.4. Some Protease Inhibitors Such as Pefabloc SC Disrupt the Endothelial Barrier at High Concentrations

Evidence of aspartic, cysteine and serine protease expression in the melanoma conditioned media suggested that some of these enzymes, along with others, may aid melanoma extravasation at the brain endothelium. Broad-spectrum inhibitors were used to assess the involvement of a range of aspartate, cysteine, and serine proteases on melanoma mediated disruption of the endothelial barrier. In the previous section, the addition of broad-spectrum inhibitors of MMPs demonstrated that protease inhibitors alone disrupted the endothelial barrier. ECIS data showed that this disruption was mostly attributed to the paracellular component (Rb), but also seen in the basolateral component (Alpha). Hence, a variety of commercially available inhibitors were added to the brain endothelial cells (hCMVECs), and ECIS, which is the more sensitive barrier impedance sensor [48], was used to establish a non-toxic, working concentration range for all the subsequent melanoma inhibition assays. Nine protease inhibitors (Table 5) were added to the hCMVECs, at concentration ranges within or above that recommended by the supplier and based on the literature. Figure 5 shows that the inhibitors E-64, leupeptin, pepstatin and phosphoramidon alone had no effect on the endothelial barrier and were used at the top concentration for further inhibition experiments. EDTA alone did not decrease the barrier resistance for 30 h post treatment, but past this timepoint it decreased the barrier resistance significantly (Appendix A). Antipain and bestatin disrupted the barrier resistance at the top concentration, whereas Pefabloc SC was toxic at both 1 mg/mL and 100 µg/mL. Aprotinin was the only inhibitor that mildly, but significantly, increased the barrier resistance. Inhibitors which disrupted the endothelial barrier on their own were evaluated and a lower appropriate working concentration was established as detailed in Table 5 along with their specificities.

### 3.5. Treatment of Melanoma Cells with a Range of Proteases Inhibitors Does Not Prevent Melanoma Mediated Brain Endothelial Disruption

Inhibitors of appropriate concentrations which were not toxic to the brain endothelial cells were incubated with three different melanoma lines individually. The melanoma-inhibitor complex was then added to the brain endothelial monolayer to assess the change in barrier resistance. Figure 6 and Figure 7 show the ECIS traces of the melanoma line NZM7. Contrary to the hypothesis, most of the protease inhibitors did not inhibit the melanoma mediated disruption of the endothelial barrier. Conversely, several inhibitors seemed to facilitate melanoma mediated barrier disruption as they enhanced the extent of the barrier disruption (Figure 6E–H). These changes also occurred past 4 h of treatment, well after the initial and predominant insult. This data was replicated in all three melanoma lines where the addition of an inhibitor did not impede their ability to disrupt the endothelial barrier (Figure 7). Only with a higher concentration of Pefabloc SC (49.75 µg/mL) was there a visible difference in resistance, but only for the NZM74 line (Figure 7). Notably, the inhibitor alone (green) also showed a transient increase in barrier resistance for this example (Appendix A), suggesting an inhibitor-only effect rather than a melanoma-protease effect. This data was only seen for one melanoma line and was, therefore, irreproducible.

## 4. Discussion

Herein, the involvement of proteases in melanoma mediated disruption of the brain endothelial cell barrier was assessed. The hypothesis was that various proteolytic enzymes, such as MMPs and serine-, cysteine- and aspartic- proteases, expressed by the melanoma lines facilitated the breakdown of junctional molecules to initiate disruption of the endothelial barrier. The assessed melanoma lines showed an expression of various proteases including MMP-2, Cathepsin B and Cathepsin D. MMPs -3 and -9, which are commonly correlated with melanoma metastases, were not detected in melanoma conditioned media, though these have conflicting roles in melanoma progression, with data that suggest both pro and anti-tumour activity [30,59,60,61]. Surprisingly, the urokinase plasminogen activator (uPA) was not detected in the conditioned media, although previous studies have shown that uPA mRNA is upregulated in metastatic cancer over benign nevi [44]; however, the receptor uPAR was detected. uPAR is a co-factor for plasminogen activation and interacts with uPA to cleave plasminogen into its active form, plasmin. Plasmin is a serine protease that cleaves fibrin, functioning as a de-clotting agent. Interestingly, uPAR is typically a cell-surface molecule [62] but can be cleaved in monocytes to act as a biological activator of chemotaxis and cell-adhesion [63], which may explain its expression in melanoma conditioned media. 

The addition of MMP-2 to the apical face of the brain endothelial monolayer showed no effect on the overall endothelial barrier function. The addition of broad-spectrum inhibitors to the melanoma cells also did not hinder the ability of the melanoma lines to disrupt the endothelial barrier and similar results were seen with a range of inhibitors of other proteases. An important point to note, was that the protease inhibitors were co-incubated with the melanoma cells and added to the endothelial cells as a melanoma-inhibitor cocktail. This was performed to ensure that the inhibitors were not only present on the melanoma alone but also at the invasive front on the apical surface of the endothelial cells, upon and after melanoma addition. The inhibitors were, therefore, expected to block some protease activity if it were majorly involved in mediating endothelial barrier disruption. The results indicate that this was not the case for any of the melanoma lines assessed. 

Intriguingly, the addition of some inhibitors such as batimastat, marimastat and Pefabloc SC, disrupted the endothelial barrier independent of the melanoma cells, which highlights that the inhibitors have an effect (which is potentially detrimental) on the endothelial barrier alone. Interpreting the specificities of the MMP inhibitors showed that there was no overlap of MMP inhibition specificity, exclusively between marimastat and batimastat; however, there were four MMPs (MMP-1, -2, -7 and -9), which were inhibited by all three inhibitors (Appendix A). Interestingly, batimastat was found to be an extremely potent inhibitor of MMP-1 (IC_50_: 3 nM, as described by the manufacturer, R&D Systems). Marimastat was also potent but less than the batimastat, whereas ONO4817, with an IC_50_ value of 1600 nM, was the least potent. This trend is depicted in Table 3, and also matched the barrier disruption trends in Figure 3 and Appendix A, where the inhibitor batimastat disrupted the barrier the most. This, however, is not conclusive nor been tested in this paper, and it is likely that the high concentrations of drugs, in general, have a detrimental effect on the brain endothelial barrier. The effect, however, was not as large as what we see with the melanoma lines [5], or with inflammatory cytokines [48].

Cumulatively, these results suggested that although several proteases may affect the endothelial cells independently over time, the MMPs and other proteases do not make a reliable target for blocking melanoma metastasis at the locus of extravasation, which is at the apical face of the endothelial cells. It also suggests that MMPs likely do not influence the melanoma mediated endothelial barrier disruption we have previously seen in [5]. These results were initially unexpected, as proteases have proven to facilitate melanoma metastases through the ECM at the primary tumour site to facilitate travel to the nearest blood vessel for intravasation into the circulation [9,64]. Melanoma protease expression is also correlated with a better chance of extravasation at the secondary site [65] and there is corresponding evidence of melanoma migration through the endothelial monolayer being inhibited by protease inhibitors [35]. In-depth investigation of the literature on protease mediated migration suggests that although proteases play an important role in melanoma migration through the ECM at the primary site, at the secondary site, proteases may only facilitate migration after the extravasation step which occurs past the endothelial monolayer and in the basement membrane and parenchyma [66], depicted in Figure 8. This was deduced as most migration studies use Matrigel based scaffolds to assess invasion rather than cellular barriers, such as the vascular endothelium. In vivo studies suggest that proteases may also be important in regulating clotting agents and platelet interactions which are major support factors for circulating melanoma cells; therefore, it is likely that the correlation between a high expression of proteases and metastatic potential is attributed to (i) traversal through ECM at the primary site, (ii) survival in the circulation and (iii) traversal through the basement membrane after the more profound layers of the blood–brain barrier have been breached, but not attributed to extravasation at the site of the brain endothelial junctions which need to be disrupted first.

Additionally, it is important to establish reliable and relevant models when designing an experimental question. It can be proposed that there are differences in the cell models grown on flat surfaces and across Transwell systems as used by Fazakas, Wilhelm [35] and this leads to disparities between cross-modal analyses. For this paper, high-throughput, real-time impedance biosensing was used to temporally assess the fast-acting mechanism occurring at the invasive front of melanoma extravasation. This front is critically on the apical face of the highly polarised brain endothelial cells, interfacing the blood. The sensitivity of impedance sensing allowed for the detection of very small changes occurring during the melanoma–endothelial interaction, and the addition of all our treatments to only the apical face of the endothelial cells allowed for spatial assessment of the effect of the proteases on the endothelial monolayer.

This was important as it revealed that the inhibition of proteases could not hinder melanoma mediated disruption of the endothelial barrier junctions. Although MMPs and other proteases may play an important role in aiding melanoma invasiveness past the endothelium, melanoma cells must still disrupt the endothelial barrier first to traverse through the paracellular space; therefore, future studies will assess the expression and effect of pre-existing key molecular players, present at the invasive front of the melanoma cells. This is particularly important due to the fast nature of the melanoma effect. If a link is found, studies must translate this research into sheer-based systems of a complete BBB, to better replicate the apical front of the brain endothelial barrier. 

Brain-metastatic cancers have poor clinical outcomes [67,68,69,70] and are substantially involved in cancer treatment failure [71,72]. This provides the biological rationale to target metastasis in the initial process of extravasation, at the brain endothelium. Thereby, we need to identify and potentially block the metastatic mechanisms used by cancer cells, whilst they are still an accessible target in the blood. Consequently, there is an urgent need to understand the mechanisms of extravasation that can have a therapeutic role in brain metastases. In this paper, we propose that proteinases such as MMPs are not primarily involved in the melanoma mediated barrier disruption of brain endothelial cells and, therefore, do not make reliable targets for therapeutic intervention at the brain endothelial front. 

## Figures and Tables

**Figure 2 biosensors-12-00660-f002:**
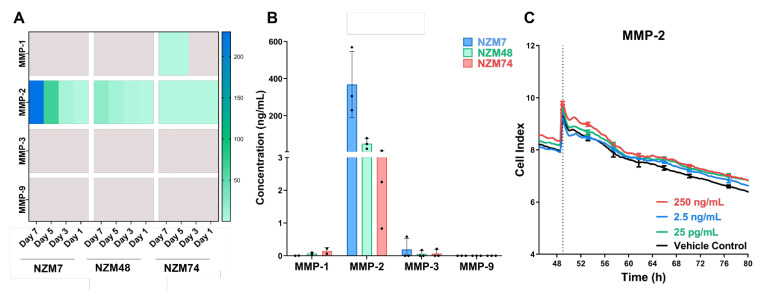
(**A**) Quantitative MMP expression as detected by Luminex, in melanoma conditioned media over time, represented as a heatmap. (**B**) Quantitative “maximum” expression of MMPs as detected by Luminex in melanoma conditioned media collected from Day 7. Black dots represent independent experimental repeats (**C**) Effect of highly expressed MMP-2 on brain endothelial monolayer.

**Figure 3 biosensors-12-00660-f003:**
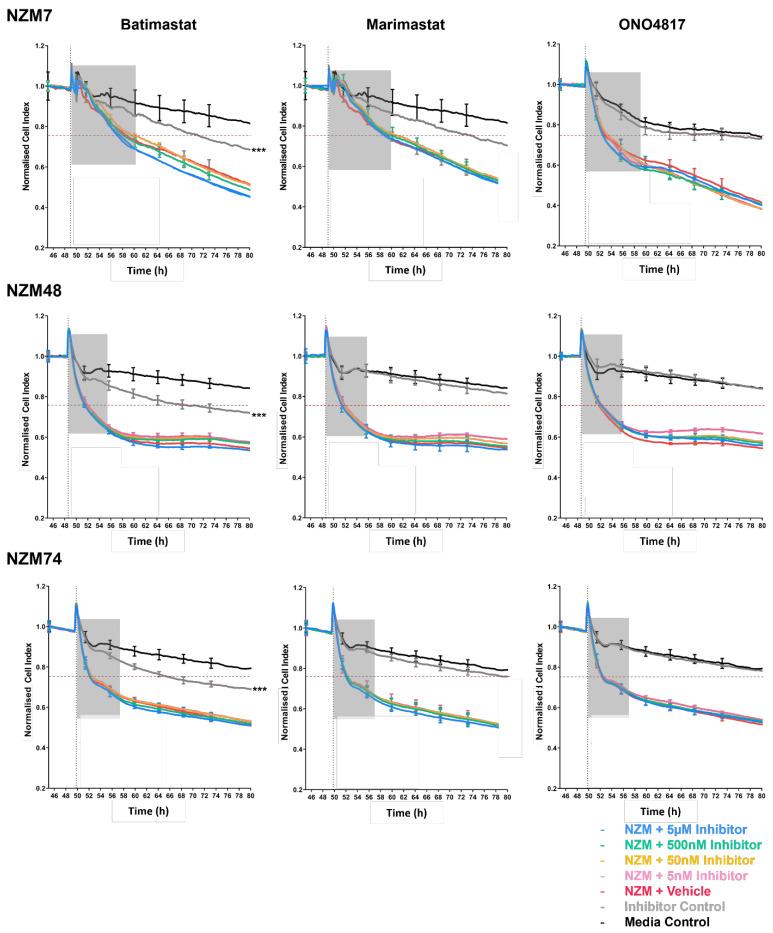
Normalized Cell Index as measured by xCELLigence (at 10,000 Hz) of hCMVECs over time after addition of NZM7 with three different broad-spectrum inhibitors, namely, batimastat, marimastat and ONO4817. Cells were added (dotted line) at an Effector:Target (E:T) ratio of 1:1 where 1 NZM cell was added for 1 endothelial cell. Inhibitors were added a top concentration of 5 µM in a series dilution of 1:10. Data show the mean ± SD (*n* = 3 wells, except NZM7- Batimastat-grey and Marimastat-grey, where *n* = 1 well is displayed due to electrode destabilisation) from 1 experiment which is representative of at least 2 independent experiments. Grey boxes show the time frame by which we expect to start seeing a change in effect if the melanoma protease were majorly involved. This is at the initial phase of the melanoma insult which typically occurs within the first few hours of addition. Red horizontal line shows the arbitrary cut-off by which we expect an improvement in CI if MMPs were majorly involved in the initial disruption caused by melanoma cells. An 80 h endpoint for at least 2 independent experiments was compared relative to their appropriate controls using a two-way ANOVA with Tukey’s range test *** *p* < 0.001). Control for Inhibitor control was Media Control. Control for NZM + Inhibitor was NZM + Vehicle Control. The endpoint represents the maximum number of hours recorded to quantify any treatment-based effect that is maintained.

**Figure 4 biosensors-12-00660-f004:**
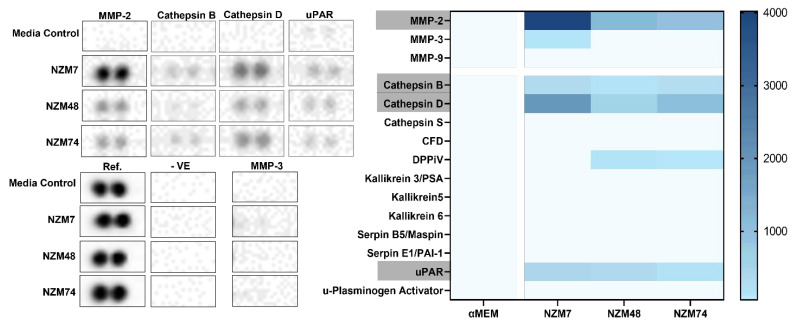
Semi-quantitative expression of proteases in melanoma conditioned media as measured by the proteome profiler. MMPs-2, -3 and -9 are added, to compare with Luminex results from Figure 1. Duplicate blots of expressed proteases are shown (**left**). MMP-3 blots are also added as an example of an undetected protein. Protein blot grey-scale intensities were recorded. Duplicates were averaged and the negative reference was subtracted from each blot. The αMEM-media control intensity for each protein blot was subtracted from the melanoma treatments and this value is presented in the heat map (**right**). Difference was used instead of Ratio or Fold Change as several media controls gave 0 as a reading. making the ratio non-sensical and unreliable. Protease expressions are represented in a blue scale with the smallest intensity value of 50 (after negative and control adjustments).

**Figure 5 biosensors-12-00660-f005:**
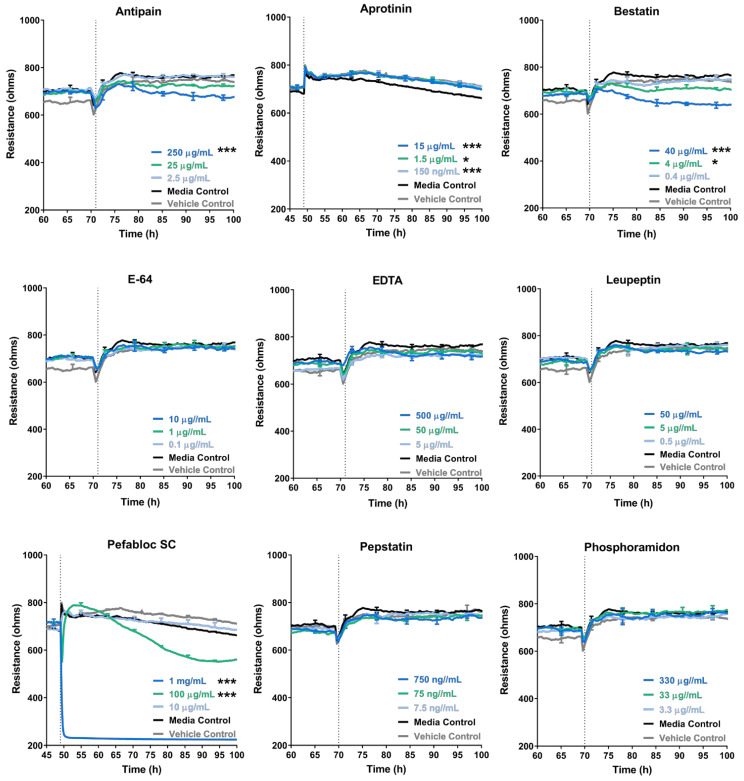
Effect of protease inhibitors on brain endothelial barrier resistance. Unmodelled resistance (at 4000 Hz) of hCMVECs over time after the addition of nine different protease inhibitor. Inhibitors were added (dotted line) at top concentrations as recommended by the supplier and the literature in a dilution series of 1:10. Data show the mean ± SD (*n* = 3 wells) from 1 experiment which is representative of 2 independent experiments. A 100 h endpoint for 2 independent experiments was compared relative to their vehicle controls using a two-way ANOVA with a Tukey’s range test (* *p* < 0.05and *** *p* < 0.001).

**Figure 6 biosensors-12-00660-f006:**
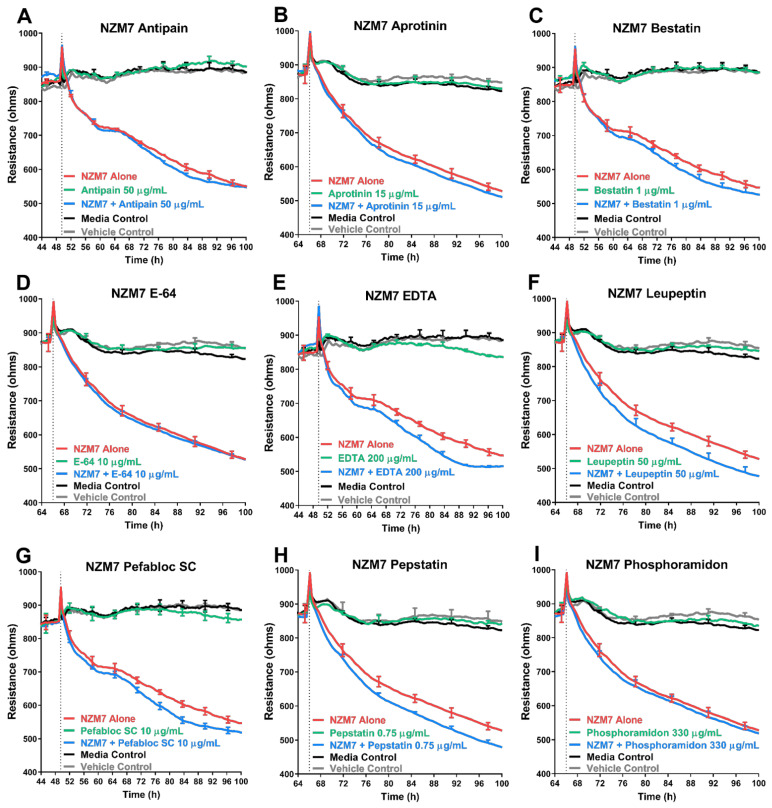
Effect of protease inhibitors on NZM7 mediated disruption of brain endothelial barrier resistance. (**A**–**I**) Unmodelled resistance (at 4000 Hz) of hCMVECs over time after addition of NZM7 with nine different protease inhibitors. Cells were added (dotted line) at an Effector:Target (E:T) ratio of 1:1 where 1 NZM cell was added for 1 endothelial cell. Inhibitors were added at relevant non-toxic top concentrations. Data show the mean ± SD (*n* = 3 wells) from 1 experiment. Similar results were seen for two other melanoma lines.

**Figure 7 biosensors-12-00660-f007:**
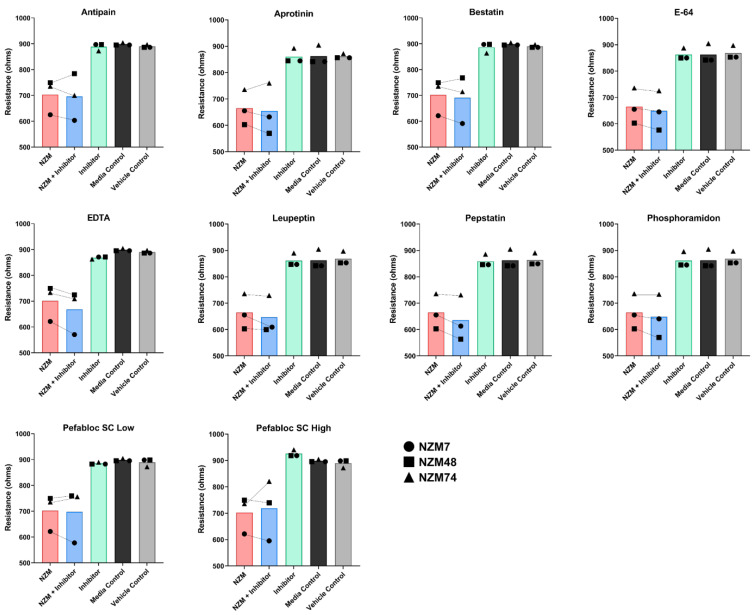
Comparable effect of protease inhibitors on the ability of three NZM lines to mediate disruption of brain endothelial barrier resistance. Resistances measured at 80 h are displayed, as by this timepoint, the major response was completed. Circle: NZM7, Square: NZM48, and Triangle: NZM74. Pefabloc SC low was at 10 µg/mL, Pefabloc SC High was at 49.75 µg/mL. Green, Black and Grey bars show the addition of different controls to the hCMVECs. Red and Blue bars show the treatment groups. An upward slope from NZM (Red) to NZM + Inhibitor (Blue) suggests that protease inhibition impedes the ability of NZM cells to disrupt the endothelial barrier, thereby giving a higher barrier resistance. Notable differences are that which change the resistance by at least 100 Ohms which showcase an over 10% improvement in barrier resistance.

**Figure 8 biosensors-12-00660-f008:**
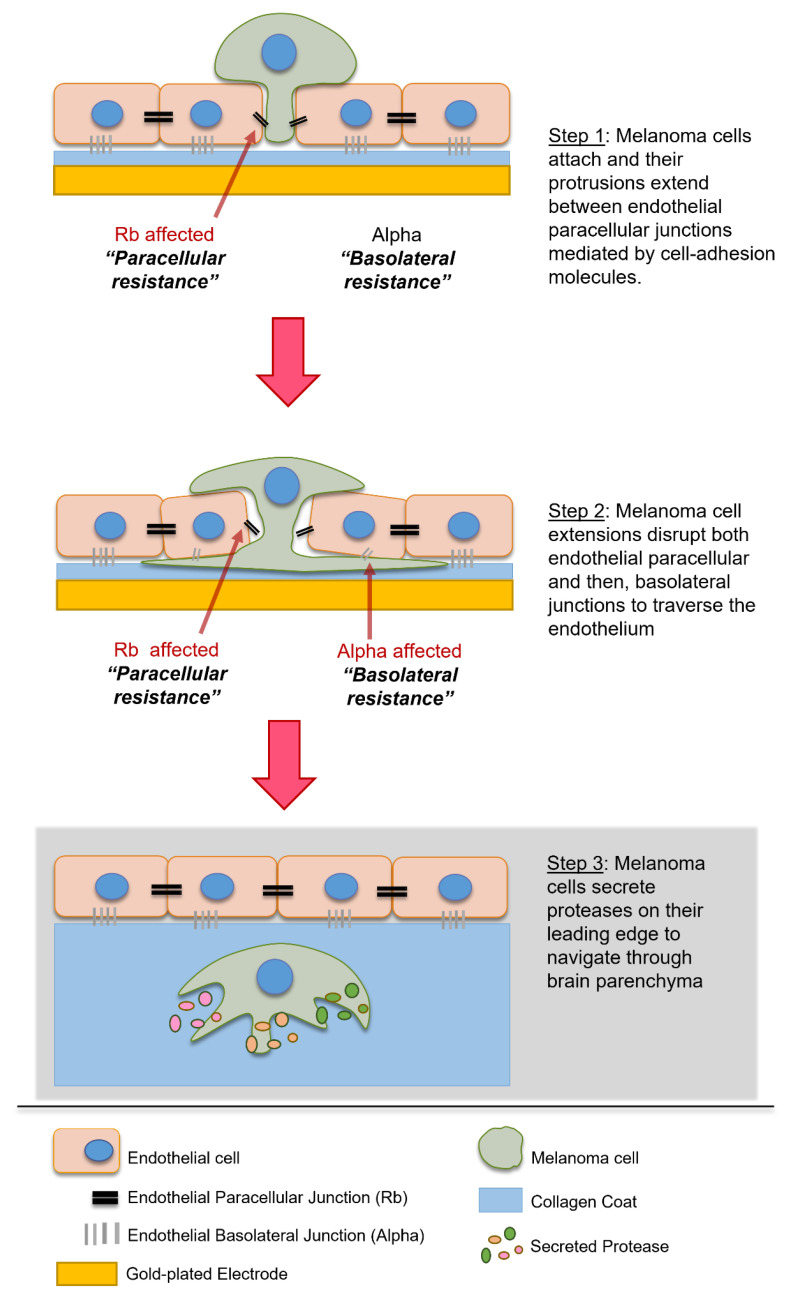
Schematic showing interpretation of results. Grey section suggests proposed mechanism of MMPs and proteases in metastasis as per the literature, due to a lack of evidence of their influence in endothelial extravasation.

**Table 1 biosensors-12-00660-t001:** List of human-derived New Zealand melanoma (NZM) lines.

Melanoma Line	Research Resource ID
NZM7	CVCL_D843
NZM48	CVCL_S423
NZM74	CVCL_0D38

**Table 2 biosensors-12-00660-t002:** Protease-based treatment of melanoma addition to the brain endothelial cells.

Assay	Inhibitor	Starting Concentration	Concentration at Melanoma Treatment	Final Concentration
xCELLigence	Batimastat	20 μM–20 nM	10 μM–10 nM	5 μM–5 nM
	Marimastat	20 μM–20 nM	10 μM–10 nM	5 μM–5 nM
	ONO4817	20 μM–20 nM	10 μM–10 nM	5 μM–5 nM
ECIS	Antipain dihydrochloride	200 µg/mL	100 µg/mL	50 µg/mL
	Aprotinin	60 µg/mL	30 µg/mL	15 µg/mL
	Bestatin	4 µg/mL	2 µg/mL	1 µg/mL
	E-64	40 µg/mL	20 µg/mL	10 µg/mL
	EDTA	800 µg/mL	400 µg/mL	200 µg/mL
	Leupeptin	200 µg/mL	100 µg/mL	50 µg/mL
	Pefabloc SC	40 µg/mL	20 µg/mL	10 µg/mL
	Pepstatin	3 µg/mL	1.5 µg/mL	0.75 µg/mL
	Phosphoramidon	1320 µg/mL	660 µg/mL	330 µg/mL

**Table 3 biosensors-12-00660-t003:** Specificity of broad-spectrum MMP inhibitors sorted by potency as defined by the manufacturer, R&D Systems.

Batimastat	IC_50_: nM	Marimastat	IC_50_: nM	ONO4817	Ki (IC_50_: nM for MMP-1)
MMP-1	3	MMP-9	3	MMP-12	0.45
MMP-2 *	4 *	MMP-1	5	MMP-2 *	0.73 *
MMP-9	4	MMP-2 *	6 *	MMP-8	1.1
MMP-7	6	MMP-14	9	MMP-13	1.1
MMP-3	20	MMP-7	13	MMP-9	2.1
				MMP-3	42
				MMP-7	2500
				MMP-1	1600

**Table 4 biosensors-12-00660-t004:** Expression of a range of proteases in melanoma conditioned media collected at Day 7.

Assessed Proteases		Mean Pixel Intensity				
Protease Name	Protease Type	αMEM	NZM7	NZM48	NZM74	Present
Cathepsin B	Cysteine Protease		320.814	128.485	247.203	Yes
Cathepsin D	Aspartic Protease	35.95	1908.8495	556.6415	1020.4705	Yes
Cathepsin S	Cysteine Protease	54.571	88.5355	64.742	95.6315	No
Complement Factor D	Serine Protease					No
DPPiV *	Serine Protease			147.778	61.0355	Yes *
Kallikrein 5	Serine Protease					No
Kallikrein 3/PSA	Serine Protease					No
Kallikrein 6	Serine Protease					No
uPAR	Involved in Plasmin Activation	99.339	475.228	423.4705	255.607	Yes
uPA	Plasmin Activator					No
Serpin B5/Maspin	Endopeptidase Regulator					No
Serpin E1/PAI-1	Serine Protease Inhibitor (e.g., uPA)		53.121		30.3285	No

* Detected at very low levels but over the threshold intensity cut-off of 50 over media control in two melanoma lines.

**Table 5 biosensors-12-00660-t005:** Specificity of protease inhibitors and established working concentrations for inhibition assays.

Assessed Protease Inhibitors		Recommended Concentration		
Inhibitor	Specificity	Concentration Tested (µg/mL)	Concentration Used (µg/mL)	Toxic to Barrier
Antipain dihydrochloride	Broad-spectrum	250	50	Yes
Aprotinin	Serine Protease	15	15	No
Bestatin	Amino Peptidase	40	1	Yes
E-64	Cysteine Protease	10	10	No
EDTA	Metallo Protease	500	200	Yes
Leupeptin	Serine, Cysteine Protease	50	50	No
Pefabloc SC	Serine Protease	1000	10	Yes
Pepstatin	Aspartic Protease	0.75	0.75	No
Phosphoramidon	Metallo-endopeptidase	330	330	No

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
