# Peer review of "Melanoma Mediated Disruption of Brain Endothelial Barrier Integrity Is Not Prevented by the Inhibition of Matrix Metalloproteinases and Proteases"

_biosensors, 2022, doi:10.3390/bios12080660_

Round 1
Reviewer 1 Report
Anchan et al. used a TEER system to measure the effects of melanoma cell lines on the integrities of BBB and screened an array of protease inhibitors to recover the detrimental impact of melanoma. The idea of this manuscript is novel and interesting, and the authors have provided sufficient data to support it. However, there are quite a few issues with their methods that I am very concerned about, I would suggest the authors revisit their methods, they should use widely applied methods to confirm their hypothesis.
1. First of all, the authors trypsinized the melanoma cells and applied the cell suspension to the endothelial cells (ECs). This method to me is very problematic. Have the authors quenched the trypsin before treating the ECs? If not, all the detrimental impacts probably came from trypsin rather than the melanoma cells. To address it, I would suggest the authors treat the ECs with the non-cancer cells with the same protocol.
2. I also do not understand that, for most TEER systems, there are two chambers (that is another issue, in a sensor journal, the author did not bother to put a device design figure in the manuscript). The authors could seed the ECs on the top member, then seed the melanoma cells in the bottom chamber, providing noncontact interactions between two types of cells (there are plenty of papers that have applied this design). The current method does not mimic the pathological event and will bring lots of potential issues.
3. The authors should also provide immunofluorescence images to further confirm the formation of a high quality EC monolayer.
Author Response
Reviewer 1: Somethings to be improved.
Anchan et al. used a TEER system to measure the effects of melanoma cell lines on the integrities of BBB and screened an array of protease inhibitors to recover the detrimental impact of melanoma. The idea of this manuscript is novel and interesting, and the authors have provided sufficient data to support it. However, there are quite a few issues with their methods that I am very concerned about, I would suggest the authors revisit their methods, they should use widely applied methods to confirm their hypothesis.
Comment 1. First of all, the authors trypsinized the melanoma cells and applied the cell suspension to the endothelial cells (ECs). This method to me is very problematic. Have the authors quenched the trypsin before treating the ECs? If not, all the detrimental impacts probably came from trypsin rather than the melanoma cells. To address it, I would suggest the authors treat the ECs with the non-cancer cells with the same protocol.
Response 1: Thank you for this comment. We used TrypLE to harvest melanoma cells, which unlike trypsin is safer and better tolerated by the cells. TrypLE does not need to be quenched (for example by FBS) but can be diluted to be inactive. Regardless, to ensure we have the cleanest, consistent results, we dilute the TrypLE with media+FBS. We then centrifuge this harvest to generate our cell pellet and completely decant all TrypLE from our treatment. The melanoma cells are added TrypLE-free to the endothelium. The product details can be found here; or see reference [1] where TrypLE was shown to be relatively more tolerated by cell lines.
We have now clarified this in the methods section of our paper under “Protease Based Treatment of Melanoma Addition to Brain Endothelial Cells”.
Comment 2. I also do not understand that, for most TEER systems, there are two chambers (that is another issue, in a sensor journal, the author did not bother to put a device design figure in the manuscript). The authors could seed the ECs on the top member, then seed the melanoma cells in the bottom chamber, providing noncontact interactions between two types of cells (there are plenty of papers that have applied this design). The current method does not mimic the pathological event and will bring lots of potential issues.
Response 2: Thank you; our research focuses on melanoma metastases from the blood into the brain, through the physical interaction with the brain endothelium. The brain endothelium is a specialised and polarised cell type as in vivo, it has a side facing the blood (apical) and a side facing the brain parenchyma (basal side attached to basement membrane). Therefore, applying melanoma cells under the endothelial monolayer would not be physiologically relevant to our experiments. We aim to add melanoma cells directly on the top of the endothelium – to the apical face, which in vivo, would be the blood facing face.
A two-chamber model would allow for molecules to slowly dissolve over time and effect the endothelium from the bottom, which physiologically is inaccurate as this would not happen in an intact blood vessel. This is often an issue with assays conducted in chamber type models. Hence, we conducted our assays in a 96-well plate model, previously established by other researchers. We have previously done the same, and our papers can be found in references: [2-7].
We have previously conducted these experiments in a Transwell or two chamber system to measure actual TEER, which show that melanoma cells affect the endothelium in a 3D system also following apical addition. Though this is not pertinent to this paper, we have added the data to our methods sections for comparison. In this same methods figure, we have now added a diagram to explain the biosensors used. We have previously published this and compared the device designs, measurement and modelling, therefore cannot add it to the paper proper. The previous published article can be found here [4].
All of the paper suggested above are cited in our submission.
Comment 3. The authors should also provide immunofluorescence images to further confirm the formation of a high quality EC monolayer.
Response 3: Thank you, we have previously published with this cell line in Biosensors to show this, here: [2, 7]. These are cited more clearly in the methods section now. To make the read easier, we have also added a supplementary figure, which shows the endothelial monolayer stained for a common junctional protein CD144 (VE-Cadherin) in a 96 well-plate system.
Reviewer 2 Report
The reading of this paper left me with quite a strange feeling. It is too confusing and it lacks too many important details regarding the experiments that were conducted, how they were conducted and the results that were obtained, how they can be understood and discussed and how they actually respond to the objectives. It was impossible for me to follow what was done and why. The impedance measurements are not understadable at all, and the reasons behind the experiments did not seem logical to me to respond to the questions that were raised. Also, there is no positive control associated with an impedance-change measurement (an actual disruption by a known mechanism, with a proof that the barrier is disrupted by another method, at least). I also found very strange that the inhibitors of the actual molecules that are supposed to disrupt the endothelial barrier presented the effect expected for the disrupting molecules themselves! Are the impedance measurements really showing what is expected? Is there a direct effect of the molecules on the impedance measurements? Can this be assessed at least?
It is impossible to accept this paper as such for all of these reasons.
The paper starts nicely with a sound introduction on the interest of studying the role of melanoma-mediated disruption of brain endothelial barrier via MMPs. Then, the authours provide the methods used to assess this role: by measuring barrier impedance with 2 different techniques that are not explained clearly at all, and the results of which are then not understandable to me. After finding that melanoma cells are indeed producing some MMPS (MMP-2), they found no effect on the barrier impedance when commercial MMP-2 was added "directly to the apical face of the cells" (whatever this means...). The "effect" shown in Figure 1C (or I'd rather say the absence thereof) looked surprising to the authors and they thus decided to jump to the next experiment consisting in using inhibitors of all MMP activity. I was totally surprised by this decision and would need a detailed explanation as to why this was the most reasonable following step... They also decided to look at the combination of melanoma cells and inhibitors and found again (using a very confusing criteria) that the inhibitors are disrupting the barrier (or, decreasing the impedance, for that matter, as the barrier disruption is never shown!). This said, the claims obtained from the ECIS results (also said, with no proof, to be more sensitive) are also not supported; or I did not get how it is possible to address paracellular and basolateral barriers independently.
The end of the last section is also not only misguiding but false, in saying that proteases inhibitors could not hinder melanoma disruption of the barrier, as the paper shows the inhibitors alone disrupted it ! I was totally lost there.
Author Response
Reviewer 2: Somethings to be improved.
Comment 1: The reading of this paper left me with quite a strange feeling. It is too confusing and it lacks too many important details regarding the experiments that were conducted, how they were conducted and the results that were obtained, how they can be understood and discussed and how they actually respond to the objectives. It was impossible for me to follow what was done and why. The impedance measurements are not understadable at all, and the reasons behind the experiments did not seem logical to me to respond to the questions that were raised. Also, there is no positive control associated with an impedance-change measurement (an actual disruption by a known mechanism, with a proof that the barrier is disrupted by another method, at least).
Response 1: Thank you. This paper investigates the effect of various proteases in mediating melanoma cell disruption of brain endothelial cells, using biosensor technology. Firstly, we must address that both biosensors xCELLigence and ECIS, are now established models of testing endothelial barrier integrity as we and other researchers have published with them extensively. The purpose of this paper was to assess a biological protease effect on a pre-existing biosensor model, rather than developing a new biosensor model. These models have already been accepted in Biosensors itself and the following references cite a selected collection of them [2-8]. These are also cited in our manuscript.
To help clarify the science and understanding behind the biosensors we have now added a schematic explaining the difference between the models in the methods sections. As this was published previously in [4], we cannot add it to the results proper; but have highlighted it in the methods.
We have also added a supplementary figure showing live-series images of melanoma addition to the brain endothelium showing that the cells create holes or gaps in the monolayer supporting xCELLigence and ECIS data. We have previously shown this to be paired with melanoma attachment to the paracellular borders of the endothelial cells, published here [2]. These are cited in the manuscript proper.
Comment 2: I also found very strange that the inhibitors of the actual molecules that are supposed to disrupt the endothelial barrier presented the effect expected for the disrupting molecules themselves! Are the impedance measurements really showing what is expected? Is there a direct effect of the molecules on the impedance measurements? Can this be assessed at least?
Response 2: Yes, we added MMP-2 as it was the protease most abundantly expressed, and this did not show any direct effect alone.
It was curious that the inhibitors, some of which are drugs clinically used in treatment, had a detrimental effect on the barrier. We commented on this observation to state that inhibitors like these, which are used in treatment, may have a deleterious effect on the blood-brain barrier endothelial cells at high concentrations.
However, in this paper, our main goal was to assess if these inhibitors, inhibited protease activity of melanoma cells in a way which blocked melanoma mediated barrier disruption of the endothelial cells. From our results, we deduce that they do not. Protease biology is complex, and MMPs often act as activators of each other as described in the introduction. Furthermore, we wondered if MMPs and other proteases were released only upon co-culture of melanoma cells with the brain endothelial cells. Therefore, when we saw no effect of MMP-2 alone, we hypothesized that this method may be too simplistic, particularly if there is a cohort of proteases working together. Hence, we decided that a better approach would be to pan-block several proteases together and take a wider approach. We have now added a paragraph to the manuscript to explain this.
Comment 3: The paper starts nicely with a sound introduction on the interest of studying the role of melanoma-mediated disruption of brain endothelial barrier via MMPs. Then, the authors provide the methods used to assess this role: by measuring barrier impedance with 2 different techniques that are not explained clearly at all, and the results of which are then not understandable to me. After finding that melanoma cells are indeed producing some MMPS (MMP-2), they found no effect on the barrier impedance when commercial MMP-2 was added "directly to the apical face of the cells" (whatever this means...).
Response 3: As in Response 1, we have now added a schematic to explain the difference between the biosensors in the methods.
Directly to the apical face of the cells means that we added the melanoma cells to the endothelial front which typically faces the blood. This is extremely important for our assay, as the brain endothelium is a specialised, and polarised cell type. In vivo, it has a side facing the blood (apical) and a side facing the brain parenchyma (basal side attached to basement membrane). Therefore, applying melanoma cells on the apical face is physiologically important in our in vitro experiments. We have now included a section of explaining this in the manuscript proper in the Methods: Cell Culture -Human brain endothelial cells (hCMVECs).
Comment 4: The "effect" shown in Figure 1C (or I'd rather say the absence thereof) looked surprising to the authors and they thus decided to jump to the next experiment consisting in using inhibitors of all MMP activity. I was totally surprised by this decision and would need a detailed explanation as to why this was the most reasonable following step...
Response 4: As stated in Response 2, we decided to tackle the problem with a broader approach in case there were protease activities occurring in a co-culture system only. This explanation is added to the manuscript.
Comment 5: They also decided to look at the combination of melanoma cells and inhibitors and found again (using a very confusing criteria) that the inhibitors are disrupting the barrier (or, decreasing the impedance, for that matter, as the barrier disruption is never shown!). This said, the claims obtained from the ECIS results (also said, with no proof, to be more sensitive) are also not supported; or I did not get how it is possible to address paracellular and basolateral barriers independently. The end of the last section is also not only misguiding but false, in saying that proteases inhibitors could not hinder melanoma disruption of the barrier, as the paper shows the inhibitors alone disrupted it! I was totally lost there.
Response 5: Thank you, we have now simplified and clarified Figure 2 (now Figure 3) to better explain our criteria.
We have previously shown ECIS and xCELLigence detect barrier disruption and have cited these in our manuscript as per response 1. We have also added a supplementary figure to show live-cell imaging pictures which show that adding melanoma cells create gaps in the endothelium. This is previously published here [2] and videos of the same are available.
ECIS has been shown to be more sensitive in various papers but can be found on this review [9] and our paper here [4]. The sensitivity comes from the fact that ECIS records resistances over several frequencies. The frequencies sweeps allow for mathematical modelling of the data by the ECIS software, which informs on the paracellular and basolateral resistances, a technique which is not available on xCELLigence or the typical EVOM/chopstick method. Furthermore, the cells being grown directly on the electrode (+basement collagen coat) further facilitate the sensitivity. These papers are cited in the manuscript.
Lastly, the inhibitor control where an effect is seen were added WITHOUT melanoma cells. So, while the inhibitors have an undetermined effect on the brain endothelium themselves, they do not alter any activity of the melanoma cells (i.e. no difference between the melanoma + inhibitor vs the melanoma + vehicle). We included the inhibitor alone data in our results as we made this side observation in our studies.
Reviewer 3 Report
The manuscript is generally well written. My main concern is related to the electrical measurements. The authors should better explain how they differentiated the paracellular form the basolateral resistence. Usually frequency sweeps are used to distinguish between paracellular and transcellular impedance (see for instance https://www.sciencedirect.com/science/article/pii/S0956566319304130) . Moreover, I don't think it is correct to refer to ECIS measurement as resistance since the instrument measure a frequency dependent impedance.
The difference in the information provided by the two electric measurement systems is also not very clear since they are very similar.
Moreover what does it means '15 Ω = considers the frequency at which the current was applied 139 (10,000 Hz)'?
Finally impedance measurment are usually normalised for the seeding area.
Reviewer 4 Report
The article ”Melanoma mediated disruption of brain endothelial barrier integrity is not prevented by inhibition of matrix metalloproteinases and proteases ” aims to demonstrate that Brain endothelial barrier integrity is not prevented by inhibiting MMP and proteases using biosensor technology.
Although the article subject of the Article is relevant, the manuscript has several points that need to be clarified.
Major point
One of the most relevant points is the difficulty of the story to be followed. The use of two different biosensors makes it challenging to interpret the results. The lack of schematic representation of each sensor is also a significant issue because it is hard to understand the assembly of the cells in the sensor and the difference between the two technologies.
Specific points:
- Please include a schematic representation of the biosensors, including the layout of the cells.
- Throughout the text, the term “Brain epithelium” (for example, line 244) is used. However, the work is performed using a cell line. This term is therefore misleading since it can be mistaken for brain organoids. Can you please clarify this issue through the text? Using “brain endothelium cell line” would be more accurate.
- Figure 2 is not easy to understand. The normalized cell index is only considered positive if at least a 25% reduction is observed, but looking at the graphs, we observed a reduction using the different inhibitor concentrations. I understand it doesn´t reach 25%., but it is not easy to interpret. Can you add a line indicating the 25% reduction, or include a table with the % of reduction for each point, at least for the last or some time points.
- In the material and methods, “The hCMVECs were seeded at 20,000 cells per well, in 100 µL of complete M199 growth media for approximately 48 hours to form a continuous monolayer, before treatment “(line 148-149). It seems that 48h is a shallow time for the cell line to reach a monolayer. It would be good to show images of the monolayer.
Round 2
Reviewer 1 Report
I do not have further comments, the authors have clarified the questions quite well.
Author Response
We thank the reviewer for their time and improvements to our paper.
Reviewer 3 Report
The authors addressed all my previous comments, improving the quality of the paper. I just believe that more literature references regarding the impedance measurement should be provided, considering previously suggested works (https://www.sciencedirect.com/science/article/pii/S0956566319304130).
Author Response
We have now added additional citations and text to the Introduction as requested. The citation we have added are as follows
- Cacopardo L, Costa J, Giusti S, Buoncompagni L, Meucci S, Corti A, et al. Real-time cellular impedance monitoring and imaging of biological barriers in a dual-flow membrane bioreactor. Biosens Bioelectron. 2019;140:111340.
- Kulczar C, Lubin KE, Lefebvre S, Miller DW, Knipp GT. Development of a direct contact astrocyte-human cerebral microvessel endothelial cells blood-brain barrier coculture model. J Pharm Pharmacol. 2017;69(12):1684-96.
- Eigenmann DE, Xue G, Kim KS, Moses AV, Hamburger M, Oufir M. Comparative study of four immortalized human brain capillary endothelial cell lines, hCMEC/D3, hBMEC, TY10, and BB19, and optimization of culture conditions, for an in vitro blood–brain barrier model for drug permeability studies. Fluids and Barriers of the CNS. 2013;10(1):33.
Reviewer 4 Report
I appreciate the changes made. This version of the article is clearer.
Author Response
We thank the reviewer for their helpful comments and review of our work.